# Study of the Steady-State Operation of a Dual-Longitudinal-Mode and Self-Biasing Laser Gyroscope

**DOI:** 10.3390/s22166300

**Published:** 2022-08-22

**Authors:** Jianning Liu, Jun Weng, Junbiao Jiang, Yujie Liu, Mingxing Jiao, Kai Zhao, Yi Zheng

**Affiliations:** 1School of Mechanical and Precision Instrument Engineering, Xi′an University of Technology, Xi′an 710048, China; 2Xi’an Institute of Modern Control Technology, Xi’an 710065, China; 3Wingtech Mobile Communications Co., Ltd., Xi’an 710000, China

**Keywords:** laser gyro, self-biasing, dual-longitudinal mode, frequency stabilization, frequency coupling

## Abstract

In order to stabilize the self-biasing state of a laser gyroscope, a dual-longitudinal-mode asymmetric frequency stabilization technique was studied. The special frequency stabilization is based on the accurate control of the intensity tuning curve in the prism ring laser. In this study, the effects of the ratio of the Ne isotopes, the inflation pressure, and the frequencies coupling on the intensity tuning curve in a laser gyro were examined. The profiles of the intensity tuning curve were simulated under the mixing ratios of Ne^20^ and Ne^22^ of 1:1 and 7:3, and the inflation pressures were 350 Pa, 400 Pa, and 450 Pa. The mixing ratio of Ne^20^ and Ne^27^ was dealt with similarly. The method for precisely adjusting the profiles of the intensity tuning curve was analyzed. The profiles were verified by experiments under different isotope ratios and pressures. Finally, based on a prism ring laser with an optical length of 0.47 m, the proposed frequency stabilization method was preliminarily verified.

## 1. Introduction

A laser gyroscope is one of the core components in an inertial guidance and navigation system [1,2]. It has the advantages of a short startup time, a large dynamic range, and a stable scale factor, etc. [3]. The ring laser is the core of the laser gyro. In order to maintain the narrow band, low power consumption, and high stability, the ring laser of the laser gyro generally uses He-Ne gas as the gain medium [4]. Experimental results show that, with a prism laser gyro with a 0.47 m optical cavity length, and when the dual-longitudinal mode is operated at a special asymmetric position in the profile of the gain, the gyro can detect the celestial component of the Earth’s rotation angular velocity without difficulty, the lock-in disappears, and a self-biasing state is induced in the gyro [5,6,7]. According to prior experiments, self-biasing has the following characteristics: (1) The dual-longitudinal mode is the same linear polarization, and the frequency coupling effect is presented; (2) The frequency interval of the two longitudinal modes is about 640 MHz; (3) The oscillation intensity of the strong and weak modes satisfies a specific proportional relation, for example I_strong_:I_weak_ = 1.4:1. After the above conditions are satisfied, the self-biasing state is generated and stabilized. Due to the same polarization of the two modes, the traditional method of frequency stabilization based on polarization orthogonality is not applicable [8]. So far, we have not realized the continuous and stable operation of self-biasing in prism laser gyros. The method of controlling the profile of the intensity tuning curve in ring lasers suggests a new way to realize a special frequency stabilization and self-biasing state. Therefore, the research in this paper lays the foundation for the new self-biasing laser gyro. In addition, the profile of the intensity tuning curve, the slope on the sides of the curve, the profile stability, and other factors affect the accuracy of the frequency stabilization in the laser gyro [9]. Therefore, precise control of the profile is helpful for improving the precision of the laser gyro.

In this paper, the Lamb theory [10,11,12,13,14] and plasma dispersion functions [15,16,17] are used to study the influence on the profile of the intensity tuning curve in the ring laser. The factors considered in this study include the Ne isotopes ratio, the inflation pressures, isotope frequency splitting and coupling, among others. The physical model of the intensity tuning curve in the ring laser was established. The profiles of the intensity tuning curve were simulated under the isotope ratios of Ne^20^ and Ne^22^ of 1:1 and 7:3, with inflation pressures of 350 Pa, 400 Pa, and 450 Pa. The mixture of Ne^20^ and Ne^27^ was also analyzed theoretically. The experimental system of the laser gyro intensity tuning curve modulation was built to verify the theoretical analysis. This study establishes the foundation for the special frequency stabilization of the dual-longitudinal-mode and self-biasing laser gyro.

## 2. Prism Ring Laser

The prism ring laser is the core device in this type of laser gyro [18,19]; the specific structure is shown in Figure 1.

The prism ring laser consists of four total reflection prisms (TRPs) that constitute the ring light path. The He-Ne mixed gas is sealed in a capillary tube by the two TRPs, shown toward the bottom in Figure 1. A gas reserve volume is connected to the capillary tube filled with He-Ne gas. In the sagittal plane, at an angle to the meridian surface, a gas-filled tube is sealed, retained for changing the gain medium. According to the aeration process, in order to ensure the purity of the He-Ne gain medium, there is a first inflation, a burn-in process, a vacuum exhaust, and a second inflation that moves through to the ring cavity. The gas component and the isotope ratio of the Ne remain the same in the two inflation processes. After the first inflation, there is a week of a continuous burn-in process for the prism ring laser. Through this burn-in, the impurities in the capillary tube can be thoroughly removed. Next, the He-Ne gas in the capillary tube is removed, and the vacuum treatment is performed. Finally, the capillary tube is filled with the pure mixture of He-Ne gas again. The purpose of the double inflation is to eliminate the impure gas particles, which may be wrapped in the molecular structure of the cavity. The impure gas particles can cause the ring laser to fail due to the purity drop in the Ne-Ne gas in the long term. In addition, the getter with a support plate is placed in the gas reserve volume to absorb the gas impurities that are released from the cavity during long-term operation.

The above process ensures the purity of the gain medium. The foreign particles that are released from the cavity or micro-leaked from the external environment cause the profile distortion of the intensity tuning curve in the rated life of the prism laser gyro. The slow pollution gradually affects the frequency stabilization accuracy, the threshold value of the ring laser, the limited structure of the mode, and even the “white light” laser failure. Next, we study the relationship between the profile of the intensity tuning curve and the influence factors such as the Ne isotopic ratio and the inflation pressure.

## 3. Profile of the Intensity Tuning Curve in a Ring Laser

### 3.1. Lamb Theory and Ring Resonator

The Lamb theory uses Maxwell equations to describe the electromagnetic field and quantization to describe the matter particles. By these means, the interaction of light with matter is analyzed. The Lamb theory is suitable for studying the frequency splitting of Ne isotopes and the modulation profile of the intensity tuning caused by frequencies coupling.

According to the Lamb theory, the self-consistent equations of the ring laser are obtained [20].
(1)$$\frac{{\dot{I}}_{1}}{I_{1}}=\frac{c}{L}\left(\alpha_{1}-\beta_{1}I_{1}-\theta_{12}I_{2}\right)$$
(2)$$\frac{{\dot{I}}_{2}}{I_{2}}=\frac{c}{L}\left(\alpha_{2}-\beta_{2}I_{2}-\theta_{21}I_{1}\right)$$
where *L* is the optical length of the ring cavity; *α* is the single round gain coefficient; *β* is the self-saturation coefficient; *θ* is the mutual saturation coefficient; the subscripts “1” and “2” represent the opposite directions of the travelling waves; *I* is the dimensionless intensity; and $\dot{I}$ is the variation in the intensity over time. The modes’ coupling effects are indicated by Lamb coefficients.

Considering that the ring laser oscillates in a steady state, ${\dot{I}}_{1}={\dot{I}}_{2}=0$. The average intensity of the ring laser is obtained by the self-consistent Equations (1) and (2),
(3)$$I=\frac{ℜ\alpha }{\beta +ℜ\theta }$$
where *I* is the average intensity, $I=\left(I_{1}+I_{2}\right)/2$; α is the average of the gain coefficient, $\alpha =\left(\alpha_{1}+\alpha_{2}\right)/2$; and *β* is the average of the self-saturation coefficient, $\beta =\left(\beta_{1}+\beta_{2}\right)/2$. According to the third-order perturbation theory, $\theta =\theta_{12}=\theta_{21}$. ℜ is the stimulated radiation efficiency, and its value is related to the pressure of the He-Ne medium inflation. The relation between the ring laser intensity and Lamb coefficients is shown in Equation (3). The light intensity possesses a definite relationship with the oscillation frequency. I(ν)–ν is defined as the ring laser intensity tuning curve. This curve is the basis for the laser frequency stabilization. Both sides of the curve directly affect the frequency stabilization accuracy. Next, we continue to analyze Equation (3) by the plasma dispersion functions.

### 3.2. Dispersion Functions Theory of Plasma

The plasma dispersion functions are used to describe the plasma state of gain medium particles. If the ring laser operates under a stable state, Lamb theory can be used to study the time-varying characteristics of the intensity, which is affected by the frequency coupling effect between the reverse-travelling wave pairs. Further, the Lamb coefficients in Equation (3) are closely related to the frequency coupling effect, Doppler broadening caused by the thermal motion of atoms, and the inhomogeneous broadening of the gas. Hence, the Lamb coefficient can be expressed by the plasma dispersion functions. One form of the plasma dispersion functions is written as
(4)$$Z\left(\xi ,\eta \right)=\frac{i}{\eta \sqrt{\pi }}\int_{-\infty }^{\infty }\left[\exp\left(-y^{2}\right)\right]\left[1+i\frac{y+\xi }{\xi }\right]L\left(y+\xi \right)dy$$
where *ξ* is the frequency parameter, *η* is the ratio of homogeneous broadening to inhomogeneous broadening, and L is a Lorentz function.

The real and imaginary parts of Equation (4) are expressed, respectively, as
(5)$$Z_{r}\left(\xi ,\eta \right)=\frac{-1}{\eta \sqrt{\pi }}\int_{-\infty }^{\infty }\left[\exp\left(-y^{2}\right)\right]L\left(y+\xi \right)\left(\frac{y+\xi }{\eta }\right)dy$$
(6)$$Z_{i}\left(\xi ,\eta \right)=\frac{1}{\eta \sqrt{\pi }}\int_{-\infty }^{\infty }\left[\exp\left(-y^{2}\right)\right]L\left(y+\xi \right)dy$$

Equations (5) and (6) are singular integral functions, hence there is no explicit function expression for them. However, with the numerical integration, the value of the plasma dispersion functions corresponding to the series of different laser oscillation frequencies can be obtained. In this way, the real and imaginary part of the plasma dispersion function and their first derivative are calculated; the results are shown in Figure 2.

Next, the Lamb coefficients are expressed by the parameters associated with the plasma dispersion function, and the equation of intensity can be solved.

### 3.3. Ne Isotopes, Inflation Pressures, and Profile of the Intensity Tuning Curve

There is about a 260 MHz frequency division between the gain curve center of Ne^20^ and Ne^22^. The composition of the Ne isotopes is complex, and the ratios are different. Due to the difference in the atomic features of Ne isotopes, there are a few differences in the stimulated radiation of the He-Ne gain medium.

We define the Doppler broadening of Ne^20^ and Ne^22^ as $ku,\;\tilde{ku}$, respectively. Since the width of the Doppler broadening is inversely proportional to the square root of the atomic mass, $ku/\tilde{ku}=\sqrt{22/20}=\sqrt{1.1}$ is obtained [19]. Hence, if the Ne^20^ and Ne^22^ double isotopes are increased in different proportions, the broadening width and peak gain are modulated.

Combined with the plasma dispersion function, the Lamb coefficients *α*, *β*, and *θ* in Equation (3) are expressed as follows [21].
(7)$$\alpha =G_{0}\left[\frac{F}{Z_{i}\left(0,\eta \right)}Z_{i}\left(\xi ,\eta \right)+\frac{\tilde{F}}{{\tilde{Z}}_{i}\left(0,\eta \right)}{\tilde{Z}}_{i}\left(\tilde{\xi },\tilde{\eta }\right)\right]-\gamma$$
(8)$$\beta =G_{0}\left[\frac{F}{Z_{i}\left(0,\eta \right)}b_{j}+\frac{\tilde{F}}{{\tilde{Z}}_{i}\left(0,\tilde{\eta }\right)}{\tilde{b}}_{j}\right]$$
(9)$$\theta =G_{0}\left[\frac{F}{Z_{i}\left(0,\eta \right)}L\left(\xi_{12},\eta \right)Z_{i}\left(\xi_{12},\eta \right)+\frac{\tilde{F}}{{\tilde{Z}}_{i}\left(0,\tilde{\eta }\right)}\tilde{L}\left({\tilde{\xi }}_{12},\tilde{\eta }\right){\tilde{Z}}_{i}\left({\tilde{\xi }}_{12},\tilde{\eta }\right)\right]$$

The gain medium is assumed to contain the double isotopes Ne^20^ and Ne^22^; *F* is the ratio of Ne^20^ in the mixture. $\tilde{F}$ is the Ne^22^ proportion, $\tilde{F}=\sqrt{1.1}\left(1-F\right)$. *η* is the ratio of the homogeneous and inhomogeneous broadening, $\tilde{\eta }=\sqrt{1.1}\eta$. *ξ* is defined as the frequency parameter, $\xi =\left(\omega -\omega_{0}\right)/ku$, $\tilde{\xi }=\sqrt{1.1}\left(\xi -{\tilde{\xi }}_{0}\right)$, $\xi_{12}=\left(\omega_{1}+\omega_{2}-2\omega_{0}\right)/ku$, and ${\tilde{\xi }}_{12}=\sqrt{1.1}\left(\xi_{12}-{\tilde{\xi }}_{0}\right)$; the subscripts “1” and “2” represent the CW and CCW directions of the light in the ring cavity. *ω* is the angular frequency of the light, and *ω*_0_ is the central angular frequency of the spectral line profile. $Z_{i}\left(\xi ,\eta \right)$ is the imaginary part of the plasma dispersion function, ${\tilde{Z}}_{i}\left(\tilde{\xi },\tilde{\eta }\right)=Z_{i}\left(\tilde{\xi },\tilde{\eta }\right)$, $b_{j}=Z_{i}\left(\xi_{j},\eta \right)-\eta {Z^{\prime }}_{r}\left(\xi_{j},\eta \right)$, and ${Z^{\prime }}_{r}\left(\xi_{j},\eta \right)$ is the first derivative of the real part of the plasma dispersion function, ${\tilde{b}}_{j}={\tilde{Z}}_{i}\left({\tilde{\xi }}_{j},\tilde{\eta }\right)-\tilde{\eta }{Z^{\prime }}_{r}\left({\tilde{\xi }}_{j},\tilde{\eta }\right)$. $L\left(\xi_{12},\eta \right)$ is the Lorentz function, $L\left(\xi_{12},\eta \right)=\eta^{2}/\left(\eta^{2}+\xi^{2}\right)$. *G*_0_ is the peak gain of Ne^20^. *γ* is the loss of the light per pass, $\gamma =G_{m}/k_{0}$, *G_m_* is the peak gain of the double isotopes, and *k*_0_ is the ratio of gain to loss.

Then Equations (7)–(9) can be substituted into Equation (3):(10)$$I\left(\xi ,\eta \right)=\frac{\left[F\frac{Z_{i}\left(\xi ,\eta \right)}{Z_{i}\left(0,\eta \right)}+\tilde{F}\frac{{\tilde{Z}}_{i}\left(\tilde{\xi },\tilde{\eta }\right)}{{\tilde{Z}}_{i}\left(0,\tilde{\eta }\right)}\right]-k_{0}{}^{-1}\frac{G_{m}}{G_{0}}}{{ℜ}^{-1}\left[\frac{F}{Z_{i}\left(0,\eta \right)}b_{j}+\frac{\tilde{F}}{{\tilde{Z}}_{i}\left(0,\tilde{\eta }\right)}{\tilde{b}}_{j}\right]+\left[\frac{F}{Z_{i}\left(0,\eta \right)}L\left(\xi_{12},\eta \right)Z_{i}\left(\xi_{12},\eta \right)+\frac{\tilde{F}}{{\tilde{Z}}_{i}\left(0,\tilde{\eta }\right)}\tilde{L}\left({\tilde{\xi }}_{12},\tilde{\eta }\right){\tilde{Z}}_{i}\left({\tilde{\xi }}_{12},\tilde{\eta }\right)\right]}$$

The profile of the intensity tuning curve in the ring laser can be expressed by Equation (10).

Further, the dispersion functions and their first derivatives can be substituted into Equation (10), and the numerical relation of the intensity tuning with *ξ* and *η* is obtained, where *ξ* is related to the multiple isotopic compositions and ratios, and *η* is related to the inflation pressure. The calculation of the intensity tuning curve in the ring laser gyro is shown in Figure 3. The inflation pressure of the He-Ne mixture is assumed as 400 Pa.

As Figure 3 shows, the Ne isotope ratio of the He-Ne gain medium has a modulating effect on the intensity tuning curve. In Figure 3a, the isotope mixing ratio is Ne^20^:Ne^22^ = 1:1, the inflation pressure is 400 Pa, and the profile of the intensity tuning curve is slightly skewed toward the high-frequency parameter side. In Figure 3b, the isotope mixing ratio is Ne^20^:Ne^22^ = 7:3, the inflation pressure is 400 Pa, and the profile of the intensity tuning curve is near symmetry.

Furthermore, there are fourteen kinds of Ne isotopes known to exist, including Ne^17^ to Ne^30^ [22]. Among them, Ne^20^, Ne^21^, and Ne^22^ are relatively easy to obtain [23,24]; the other pure isotopes need to be separated from the mantle, air, meteorite, or other sources by some complex physical processes. By using a high-neutron-number Ne isotope, the profile of the intensity tuning curve can be further modulated. For example, the calculations are shown in Figure 4 for the mixed gas of Ne^20^ and Ne^27^ in the mixing ratios of 1:1 and 7:3, respectively.

Figure 4 shows the significant modulating effect on the intensity tuning curve. Assuming Ne^20^:Ne^27^ = 1:1 and an inflation pressure of 400 Pa, a dip appeared near the center of the intensity tuning curve. For Ne^20^:Ne^27^ = 7:3, with an inflation pressure of 400 Pa, double intensity peaks appeared, and the symmetry of the curve was conspicuously lost. In theory, by precisely controlling the isotopes’ composition, ratio, and the cavity loss, the profile of the intensity tuning curve, including the frequency gapping and intensity ratio of the two peaks, can be strictly modulated. The results provide the possibility for the dual-longitudinal-mode frequency stabilization in an asymmetric position in the ring He-Ne laser. Research has shown that a specific self-biasing characteristic is represented under this state in the prism laser gyro.

### 3.4. Inflation Pressure and Profile of Intensity Tuning Curve

The influence of the inflation pressure on the profile of the intensity tuning curve is complicated. First, it is related to the excitation of the gain medium. For example, the prism ring laser in this paper is excited by the high-frequency discharge (HFD) [25]. The HFD provides the necessary electric field strength to maintain a plasma pole with a relatively low supplied voltage, such as 5~7 V. However, the selection of the ideal inflation pressure for HFD is rigorous. The specific influencing factors related to inflation pressure include *η*, *ℜ*, *G_m_*/*G*_0_, and *k*_0_, presented in Equation (10). *η* is the ratio of the homogeneous broadening to the inhomogeneous broadening. *ℜ* represents the stimulated radiation efficiency. *G_m_*/*G*_0_ is the ratio of the peak gain of the double isotopes to the peak gain of Ne^20^. *k*_0_ is the ratio of gain to loss. The calculation parameters for Equation (10) are shown in Table 1, under the inflation pressures of 350 Pa, 400 Pa, and 450 Pa, respectively.

According to the parameters in Table 1, the simulation results of the intensity tuning curve are shown in Figure 5.

As shown in Figure 5, in the barometric range of 200 Pa, with a decrease in the inflation pressure, the intensity tuning curves present a decrease in the uniform saturation. Next, we describe the setup of the experimental system and the experimental verification.

## 4. Experimental Results

### 4.1. Experimental System

To further identify the factors affecting the gyro intensity tuning curve profile, confirmatory experiments were designed. The experimental system is illustrated in Figure 6. A cavity length controller with heat conduction was designed as the scanning mode unit. The heating wire was made of Ni-Cr alloy, and it was closely wound in the cavity length controller. The cavity length controller, capillary, and seal cover together formed a closed space. The temperature of the dry gas in the closed space was controlled by the heating wire; then, the refractive index of the dry gas was changed by heating, and the optical cavity length of the ring cavity was changed, in order to scan the longitudinal mode frequency.

From the prism ring gyro, the light was output through the combing prism and incident into the F-P cavity. The diagram of the longitudinal mode state was obtained from the F-P cavity, as shown in Figure 7a. The axis of abscissas in Figure 7a is the frequency parameter, which represents the oscillation frequency of the longitudinal mode. In order to accurately detect the profile of the intensity tuning curve, the detection scheme is shown in Figure 7b. The voltage applied to the heating wire in the cavity length controller was changed uniformly. Correspondingly, the frequency of the oscillation longitudinal mode was scanned. The intensity amplitude of the oscillation mode was modulated by the profile of the intensity tuning curve. Hence, by tracking the maximum of the amplitude, the profile of the intensity tuning curve was obtained, as shown in Figure 7b.

### 4.2. Ne Isotopes and Inflation Pressure Experiment

A prism ring resonator was selected for the experiment and repeatedly inflated. First, the isotope ratio was Ne^20^:Ne^22^ = 7:3, and the inflation pressures of the gain were 350 Pa, 400 Pa, and 450 Pa, respectively; the experimental results are shown in Figure 8a. Then the mixed gas with the natural Ne isotopes was dealt with in the same way, and the results are shown in Figure 8b.

Figure 8a shows the intensity tuning curve in the case of He:Ne = 15:1 and Ne^20^:Ne^22^ = 7:3. The profile of the intensity tuning curve was similar Gaussian. This result is basically consistent with the theoretical analysis shown in Figure 3b. The main reason for the symmetry difference between the experimental and the theoretical curve is that there may have been slight deviations in the Ne isotopes’ mixing ratio in the experiment. Figure 8b shows the intensity tuning curve under the condition of the natural Ne isotopes’ mixed gas. A dip appeared in the high frequency, and the intensity tuning curve showed a double peak overall. In addition, there was a mode competition effect between the ring laser modes near the center frequency; hence, the output intensity fluctuated within a certain range, and the profile became thicker. The dip was the product of the Ne isotopes and the cavity loss. Comparing the experimental results with the theoretical calculation results shown in Figure 4, the reasons for the differences are as follows. Since pure Ne^27^ is expensive, we used natural Ne gas, which contains various Ne isotopes to replace Ne^27^ in our experiments. Therefore, the general profile of the curve was consistent, such as the dip, the double peaks, etc.; however, there were still differences with the theoretical curves of pure Ne^27^. In general, the experimental results were consistent with the theoretical analysis.

As shown in Figure 8a, under the different inflation pressures, if the mixing ratio of Ne isotopes was strictly controlled, the profile of the intensity tuning curve remained largely stable. With the increase in pressure, the profile width was slightly widened. This was caused by the increase in the effects of radiation trapping. Ne atoms were in the excited state, the photons were emitted by spontaneous radiation, and the photons were trapped by other atoms in the ground state (resonance absorption). The process was equivalent to retaining the excited state but transferring the state to another atom. It caused the excited state to be transferred between Ne atoms, which had different velocities. Therefore, the life of the excited state was prolonged, and the profile of the intensity tuning curve was widened. The radiation-trapping effect was enhanced by the rising inflation pressure. However, under 450 Pa, the intensity fluctuated near the gain peak of the curve. This was caused by the rising pressure, which worsened the collisions between the particles in the gain medium. At 350 Pa, the gain of the laser tube was lower than that of the gain at 400 Pa.

The mixed gas of He and natural Ne was filled in the ring resonator, the inflation pressure was strictly controlled, and Figure 8b was obtained. Due to the randomness of the isotope ratio in natural Ne, there were some differences in the results after each inflation. Under 400 Pa, the profile of the curve was more stable, and the gain was medium. Under 350 Pa, the laser tube was extinguished during tuning. The experimental results coincide with the theoretical analysis.

## 5. Dual-Mode Stabilization at the Asymmetrical Position

Research shows that when dual modes (longitudinal modes or transverse modes) in a ring laser are operated in a special asymmetrical position of the intensity tuning curve, the Sagnac effect is enhanced [26,27,28]. For the prism laser gyro, in addition to the significant improvement in the sensitivity of the angle measurement, it also presents the self-biasing state. In addition, some other experimental results provide possibilities for the further improvement of the accuracy of the laser gyro. At present, the key problem affecting the application of the self-biasing effect is the frequency stabilization in the special state of the modes’ asymmetry.

When no modulation was applied to the intensity tuning curve by mechanical jitter, the frequency discrimination signals were modulated by the profile of the similar Gaussian intensity tuning curve, as shown in Figure 9a. Whether the dual modes were located in or deviated from the self-biasing frequency position, the frequency discrimination signals both lacked specificity, as shown in Figure 9b,c. If the modulation approach was taken to the intensity tuning curve, the ideal double peak of the profile was obtained. The frequency discrimination signals were generated from the dual modes in the respective intensity peak positions, as shown in Figure 9d. From Figure 9e,f, we can find that, after the profile modulation, the frequency discrimination signals were specific and suitable for the frequency stabilization. Hence, through the modulation of the intensity tuning curve profile, the stable operation of the double-longitudinal-mode and self-biasing laser gyro may be realized.

We carried out the preliminary experimental verification of the above frequency stabilization scheme. Figure 10 shows the prism ring laser used in the experiment. On the left side of Figure 10 is a traditional single-longitudinal-mode operating laser with an optical length of 0.28 m; on the right is the dual-longitudinal-mode laser, with an optical length of 0.47 m. The larger resonator used natural Ne isotopes, and the inflation pressure was 450 Pa. According to the observation of the F-P interferometer, the state of dual-longitudinal-mode asymmetric frequency stabilization lasted until the mode jump. At the temperature of 25 °C, the steady state was maintained for at least 10~15 min. During this period, the frequency discrimination signals of the laser conformed to the expectation shown in Figure 9e.

It should be noted that the technical scheme is still in the research phase. There are two main problems at present: First, the control accuracy of the profile needs to be further improved to match the strict requirement of the self-biasing. Second, the stability of the profile needs to be improved, especially for a longer operation. These problems are still under study.

## 6. Conclusions

In summary, through the precise control of the profile of the intensity tuning curve, the self-biasing state in a prism laser gyro is likely to be stabilized. The theoretical and experimental results show that the profile of the intensity tuning curve was closely related to the Ne isotopes’ composition, ratio, and inflation pressure. By controlling the above factors, the profile of the intensity tuning curve can be adjusted. This research provides a basis for realizing a special frequency stabilization, where double-longitudinal modes are stabilized at the asymmetrical position of the gain curve and the self-biasing phenomenon is presented in the laser gyro. The laser intensity tuning curve is determined by the gain and loss. In the future, we need to make more precise adjustments to the intensity tuning curve through precise control of the cavity loss. In addition, it is necessary to study how to stabilize the intensity tuning curve of the laser for a longer time under working conditions.

## Figures and Tables

**Figure 1 sensors-22-06300-f001:**
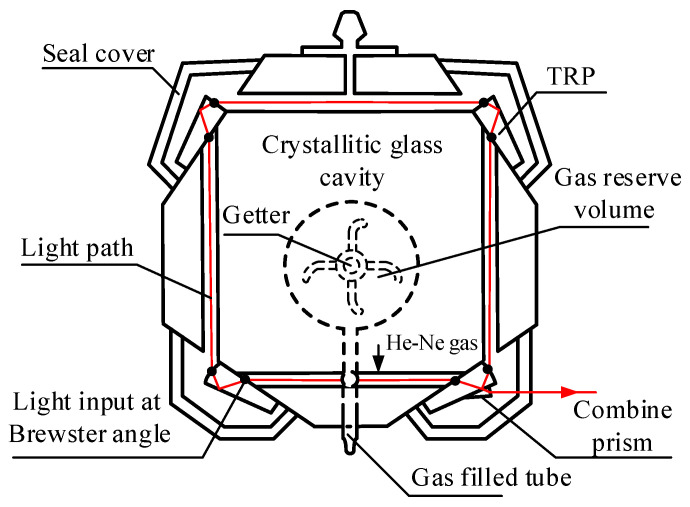
Schematic of the prism ring laser’s structure.

**Figure 2 sensors-22-06300-f002:**
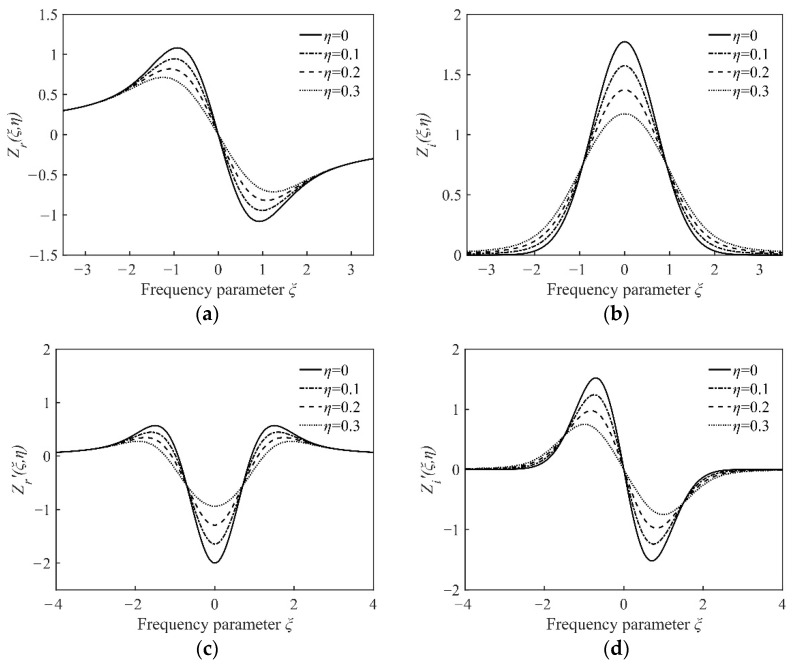
Numerical results of the plasma dispersion function. (**a**) The real part of the plasma dispersion function; (**b**) the imaginary part of the plasma dispersion function; (**c**) the first derivative of the plasma dispersion function’s real part; (**d**) the first derivative of the plasma dispersion function’s imaginary part.

**Figure 3 sensors-22-06300-f003:**
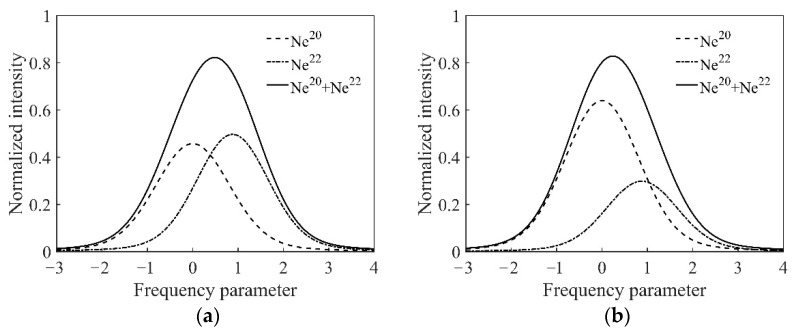
The normalized results of the intensity tuning curve in the ring laser gyro. (**a**) Ne^20^:Ne^22^ = 1:1, inflation pressure 400 Pa; (**b**) Ne^20^:Ne^22^ = 7:3, inflation pressure 400 Pa.

**Figure 4 sensors-22-06300-f004:**
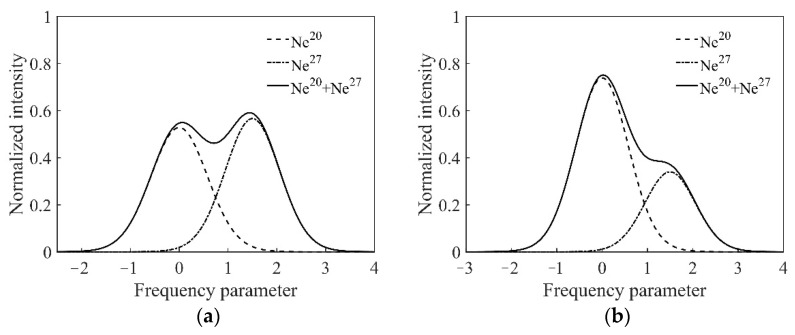
The normalized results of the intensity tuning curve in the ring laser gyro. (**a**) Ne^20^:Ne^27^ = 1:1, inflation pressure 400 Pa; (**b**) Ne^20^:Ne^27^ = 7:3, inflation pressure 400 Pa.

**Figure 5 sensors-22-06300-f005:**
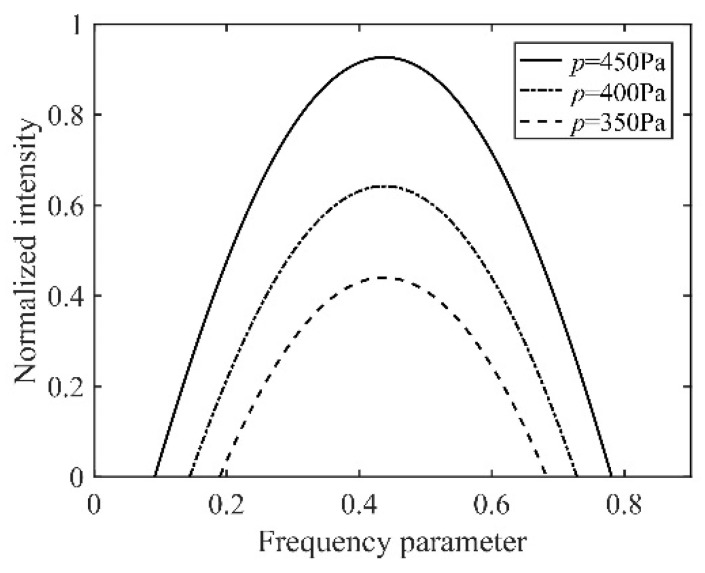
Ne^20^:Ne^22^ = 1:1, with inflation pressures of 350 Pa, 400 Pa and 450 Pa; the normalized results of the intensity tuning curve in the ring laser gyro.

**Figure 6 sensors-22-06300-f006:**
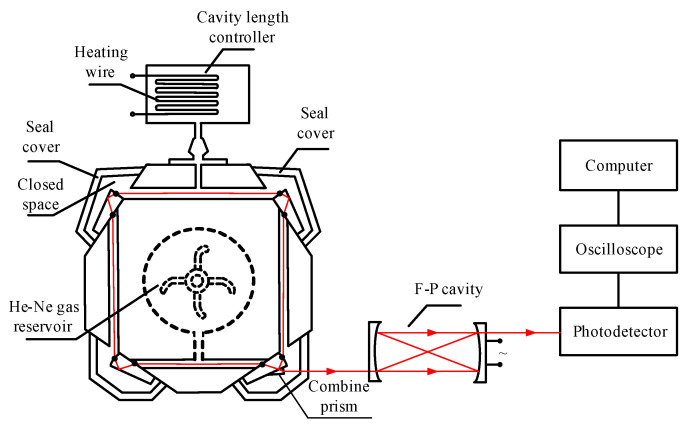
Experimental system of the prism gyro intensity tuning curve.

**Figure 7 sensors-22-06300-f007:**
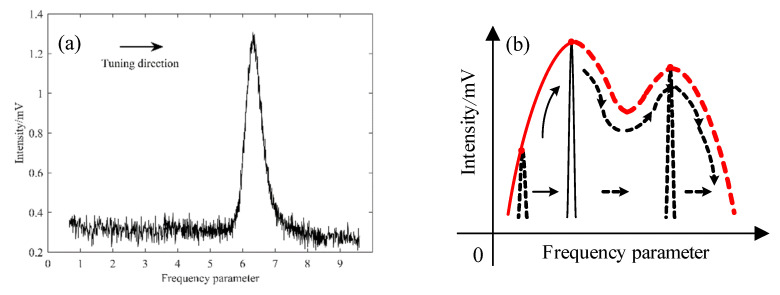
Profile of the intensity tuning curve obtained by the F-P cavity. (**a**) Frequency diagram of the longitudinal mode; (**b**) schematic of the profile detection of the intensity tuning curve.

**Figure 8 sensors-22-06300-f008:**
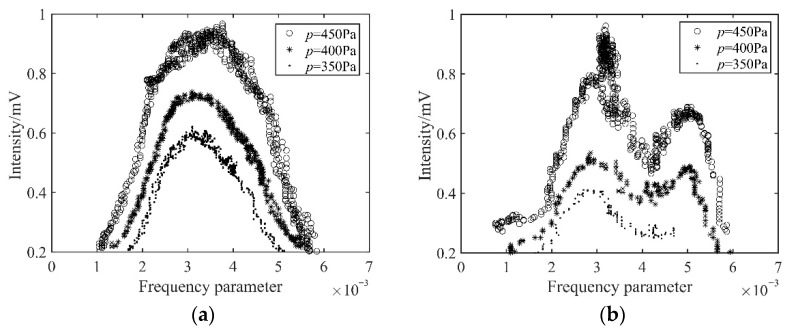
Experimental results of the profile of the intensity tuning curve in the ring gyro. (**a**) Ne^20^:Ne^22^ = 7:3, inflation pressures are 350 Pa, 400 Pa, and 450 Pa; (**b**) natural Ne isotopes, inflation pressures are 350 Pa, 400 Pa, and 450 Pa.

**Figure 9 sensors-22-06300-f009:**
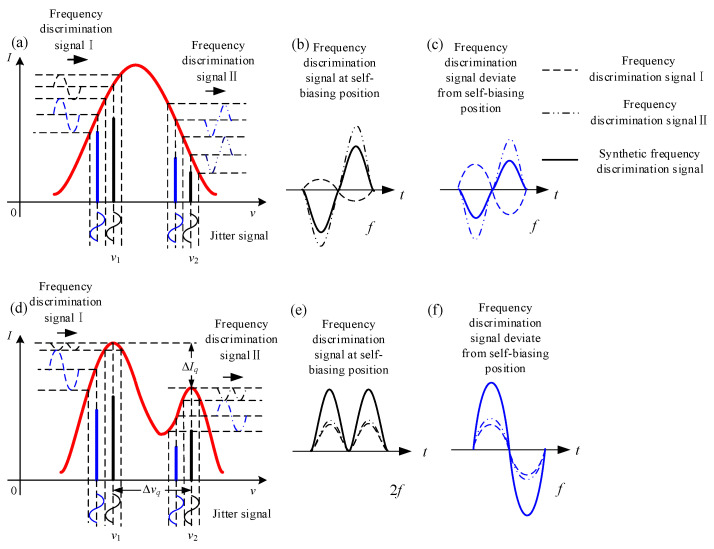
Contrast diagram of frequency stabilization under different profiles of the intensity tuning curve. (**a**) Similar Gaussian curve, with the frequency discrimination signals in the self-biasing state; (**b**) similar Gaussian curve, with the synthetic discrimination signals in the self-biasing state; (**c**) similar Gaussian curve, with the frequency discrimination signals deviating from the self-biasing position; (**d**) modulated profile, with the frequency discrimination signals in the self-biasing state; (**e**) double-peak profile, with the synthetic discrimination signals in the self-biasing state; (**f**) double-peak profile, with the frequency discrimination signals deviating from the self-biasing position.

**Figure 10 sensors-22-06300-f010:**
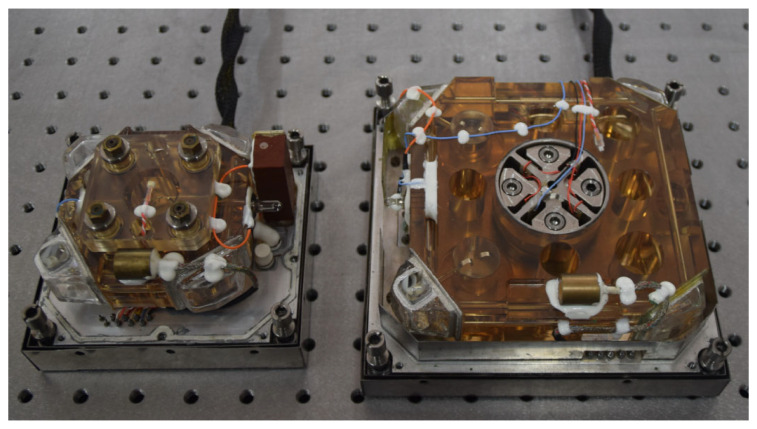
The frequency-stabilized prism ring laser.

**Table 1 sensors-22-06300-t001:** Ne^20^:Ne^22^ = 1:1, λ = 632.8 nm; calculation parameters under different inflation pressures.

Pressure (Pa)	*η*	*ℜ*	*G*_m_/*G*_0_	*k* _0_
350	0.15	0.41	0.88	1.01
400	0.20	0.42	0.87	1.02
450	0.30	0.43	0.86	1.04

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
