# Peer review of "Study of the Steady-State Operation of a Dual-Longitudinal-Mode and Self-Biasing Laser Gyroscope"

_sensors, 2022, doi:10.3390/s22166300_

Round 1
Reviewer 1 Report
J. Liu et. al. have reported how the ratio of Ne isotopes and the inflation pressure can affect the intensity tuning curve in a dual-longitudinal mode and self-biased prism ring laser gyroscope. The associated theoretical predictions are made through Lamb theory and plasma dispersion functions and further explored through experimental measurements of the intensity tuning curve. Even though this study could be of interest, especially in terms of potential applications for frequency stabilization and self-biasing effect, I cannot recommend it for publication in the current form and before making a number of clarifications and corrections.
First things first, the English of the paper should be substantially revised. Open sentences along with typos and grammatical errors have made many parts of the manuscripts such meaningless that are impossible to follow.
References or detailed derivation should be provided for Lamb coefficients in Eq. (7)-(9).
Experimental data in Fig. 8(a) and (b) should be compared with theoretical results and the reason of any discrepancy should be explained. As a minor recommendation, the quality of this figure should be improved.
The authors refer to Fig. 9 and claim that “By the modulation of the intensity tuning curve profile, the stable operation of the double longitudinal-mode and self-biasing laser gyro can be realized”. These results should be either supported by experimental observations or removed.
Author Response
Respectful reviewer:
Thank you for your professional review and the incentives for the research. We have revised the manuscript according to your comments. We believe these edits have significantly strengthened the manuscript. Please see our detailed response in the attachment. Thank you.
Jianning Liu

Reviewer 2 Report
As a referee I had difficulty to understand and follow the paper due to the English problems. There are several points to be addressed/corrected in order to evaluate the paper for publication again:
- The abstract should be improved. The purpose of the work is not clearly presented.
- The citations in the paper is problematic. A statment is made and some citations are placed like [1-4] but it is not clear how the citations are related to the statements. Please be more careful and more specific.
- Many sentences are impossible to understand, such as: Based on the experimental phenomena observed by the previous research, the prisms laser gyro with double longitudinal modes and four frequencies operating, when the double longitudinal modes satisfy the certain conditions [5]. The English is very poor and should be improved all over the paper. I am not able to follow the sentences at all. The introduction is full of such statements.
- The introduction should be extended. It should be more clear and should include more and proper citations and the aim of the study should be presented as well as the novelty with respect to existing literature. In this way it is not acceptable.
- The conclusion part should be updated. This is such a very small summary of the work without conclusions/discussions/outlook.
- Due to the English language problems I could not follow whole paper. It should be improved before consideration for publication.
Author Response
Respectful reviewer:
Thank you for your professional review. We have revised the manuscript according to your comments. We believe these edits have significantly strengthened the manuscript. Please see our detailed response in the attachment. Thank you.

Reviewer 3 Report
The paper seems good in suggesting a method to realize a self-biasing He-Ne laser gyro. Authors have used active path length control using heating coils as a dithering mechanism, which is a novel idea. The results are also encouraging. Finally, the following details need to be improved:
1. Equations (7-8): missing reference.
2. The reference [5] is non-readable for non-Russian speakers, if possible add another reference for non-Russians speakers.
Author Response
Respectful reviewer:
Thank you for your professional review and the incentives for the research. We have revised the manuscript according to your comments. Please see our detailed response in the attachment. Thank you.

Reviewer 4 Report
In this paper, the influences of the ratio of Ne isotopes, the frequency coupling and the inflation pressure on the intensity tuning curve of the prisms ring laser are studied by using Lamb theory and plasma dispersion function. The correctness of the theoretical analysis is verified by the sufficient experimental data, which provides a basis for the steady-state operation of the dual-longitudinal mode and self-biasing laser gyro. The paper is novel in content and complete in structure. In addition, the following question should be answered:
1.As one of Ne isotopes, Ne27 is not common in the traditional He-Ne gas gain media. Please explain why Ne27 is used, and the references to this point of view should be added accordingly.
Author Response
Respectful reviewer:
Thank you for your professional review and the valuable comments. We have revised the manuscript according to your comments. We believe these edits have significantly strengthened the manuscript. Please see our detailed response in the attachment. Thank you.

Round 2
Reviewer 1 Report
In this revised manuscript, the authors have provided proper response to my comments in the previous review, and they are understood well with no need for further correction or improvement. Therefore, I recommend publication of this manuscript as it is now.
Author Response
Respectful reviewer:
Thank you for your professional review again!
Jianning Liu
2022.8.16
Reviewer 2 Report
The remarks are addressed by the authors. But the paper should be checked properly before publication. The figure 6 is missing. The word formatting of equations on page 5 does not look nice.
Author Response
Respectful reviewer:
The two problems you mentioned have been revised, and the full manuscript has been checked further. Thank you for your professional review again!
Jianning Liu
2022.8.16